# An Investigation of Pro-Environmental Behaviour and Sustainable Development in Malaysia

**Mohd Yusoff Yusliza [1], Amirudin Amirudin [2], Raden Aswin Rahadi [3],
Nik Afzan Nik Sarah Athirah [1], Thurasamy Ramayah [4,5,6,7], Zikri Muhammad [1,*],
Francesca Dal Mas [8], Maurizio Massaro [9], Jumadil Saputra [1] and Safiek Mokhlis [1]**

1    Faculty of Business, Economics and Social Development, Universiti Malaysia Terengganu,
     Kuala Nerus 21030, Malaysia; yusliza@umt.edu.my (M.Y.Y.); rarahafzan@gmail.com (N.A.N.S.A.);
     jumadil.saputra@umt.edu.my (J.S.); safiek@umt.edu.my (S.M.)
2    Department of Anthropology, Faculty of Humanities, Universitas Diponegoro, Kampus Tembalang,
     Semarang 50275, Indonesia; amirudin@undip.ac.id
3    School of Business and Management, Institut Teknologi Bandung, Bandung 40132, Indonesia;
     aswin.rahadi@sbm-itb.ac.id
4    School of Management, Universiti Sains Malaysia, Gelugor 11800, Malaysia; ramayah@usm.my
5    Internet Innovation Research Center, A212, Newhuadu Business School, Minjiang University,
     200 Xiyuangong Road, Shangjie Town, Minhou County, Fuzhou 350108, China
6    Department of Management, Sunway University Business School (SUBS), 5, Jalan Universiti,
     Bandar Sunway, 47500, Malaysia
7    Faculty of Accounting and Management, Universiti Tunku Abdul Rahman (UTAR), Sungai Long Campus,
     Cheras, Kajang 43000, Malaysia
8    Lincoln International Business School, University of Lincoln, Brayford Wharf East, Lincoln,
     Lincolnshire LN5 7AT, UK; fdalmas@lincoln.ac.uk
9    Department of Management, Ca' Foscari University of Venice, Fondamenta S. Globbe, 873,
     30121 Venice, Italy; maurizio.massaro@unive.it
*    Correspondence: zikri@umt.edu.my

**Abstract:** This study aimed to examine the role of environmental commitment, environmental consciousness, green lifestyle, and green self-efficacy in influencing pro-environmental behaviour. Data were obtained through a survey of 72 students at one of the training centers in Malaysia. The hypothesized relationships were tested using partial least squares (PLS) methodology. Results showed that environmental commitment, environmental consciousness, green lifestyle, and green self-efficacy positively influenced pro-environmental behaviour, thereby providing new insights to existing literature on environmental sustainability. The results may be used by educational institutions, the government, and private agencies to strengthen students' knowledge, attitude, and behaviour that support environment-based education. The scope of the study was limited to students at a training center, so the results may not be generalizable to different settings. Another limitation was that the study used limited contextual elements. The novelty of this study is that it examined the role of environmental commitment, environmental consciousness, green lifestyle, and green self-efficacy as determinants of pro-environmental behaviour among students in an educational setting in Malaysia.

**Keywords:** sustainable development; pro-environmental behaviour; environmental commitment; environmental consciousness; green lifestyle; green self-efficacy

## 1. Introduction

### 1.1. Background

The main goal of sustainable development (SD) is to sustain a promising future and the planet. To this end, national and international authorities have issued pertinent regulations. Many researchers have been stressing the need for individuals to increase their efforts in protecting the natural environment [1,2]. The United Nations has strongly emphasized environmental protection and encouraged the whole world to maintain environmental well-being. In addition, Goal 12 of the 2030 Agenda of SD mentions the need to create awareness of SD among people around the world and promote healthy lifestyle [3]. However, despite these attempts, there have been in recent years increasing environmental issues such as environmental contaminations (air, water, and land resources), climate change, and depletion of natural resources. One of the noted major causes is human behaviours [4,5].

Due to the escalation of environmental issues attributable to human behaviours, a topic that has been dominantly investigated in environmental sustainability research is pro-environmental behaviour [5]. Pro-environmental behaviour is a set of behaviours practiced by individuals that seek to take measured actions to promote positive changes in the environment and limit the effects of human negligence [6]. Despite its importance, as Ertz et al. [7] speculated, pro-environmental behaviour may not be adopted by individuals due to some factors including time, cost, and effort. Individuals' intention to being ecologically friendly may be influenced by their beliefs, motives, and commitment to the environment [8]. Additionally, this responsible behaviour is more likely to be engaged by highly educated individuals with profound environmental knowledge and motivation [9]. Some individuals may be inspired to be involved in pro-environmental behaviour by their self-identity and biospheric values [10]. Pro-environmental behaviour is also influenced by distinctive factors, including environmental commitment [11], green lifestyle or pro-environmental lifestyle [12], self-efficacy [13,14], environmental consciousness or awareness [4,15,16], and goal-framing differences [9].

In promoting pro-environmental behaviour, colleges, universities, and training centers play an essential role, since individual behavioural change can be easily fostered among young generations [17,18]. Additionally, in the context of education, organizations are interested in having pro-environmental shifts due to their sustainability goals and implications towards student enrolment [19]. For example, students may be intrigued to immerse themselves in sustainable environmental behaviour through intensification of environmental education [5]. Within this context, several studies have explored students' pro-environmental behaviour, including food-related environmental beliefs and behaviours [20], electronic environmental knowledge [21], students' intention and loyalty towards green products [22], students' gender differences interplay in pro-environmental behaviour [23], and even the use of emoticons to encourage students to recycle [24].

Hence, to reach the sustainability goals, there is a need to foster people's personal behaviours with regards to the environment starting from a young age. Students need to engage in pro-environmental behaviour since they will be exposed to the impacts of environmental issues in the future, and they can stimulate environmental efforts [25]. In this respect, the present study took place at one of the training centers in Malaysia. The study aimed to determine whether the ecological behaviours performed by the students were derived from their own initiatives in performing pro-environmental behaviour or were prompted by the green behaviour control of the training center. Thus, the focus was on students' awareness regarding personal pro-environmental behaviour as voluntarily performed or as part of the rules and regulations set by the management to preserve the surrounding.

### 1.2. Need for the Study

This study examined the direct influence of environmental commitment, environmental consciousness, green lifestyle, and green self-efficacy on pro-environmental behaviour. It should be

highlighted that after a review of the most significant and recent works students' pro-environmental behaviour, such as [22,25–36], no similar research was found. In addition, previous empirical studies on pro-environmental behaviour have been conducted in relation to environmental commitment [37–39], environmental consciousness [40,41], lifestyle practices [42], green lifestyle [5,43], and self-efficacy [13,44–47]. However, these studies were conducted in different contexts.

In this regard, this work offers an original perspective on the relationships between environmental commitment, environmental consciousness, green lifestyle, green self-efficacy, and pro-environmental behaviour. Its novelty can be justified as follows:

- No studies, to the best of our knowledge, have so far explored these relationships.
- The existing literature on pro-environmental behaviour is largely influenced by perspectives from developed countries [19,23,26,27,32], while existing works from developing countries [31] generally do not discuss the topic in the Malaysian context.
- No studies have provided empirical evidence from Malaysia on the theoretical framework presented here. The current study fills the gap in the literature on sustainability in Malaysia.

Thus, this study aimed to answer the research question of whether environmental commitment, environmental consciousness, green lifestyle, and green self-efficacy significantly affects the practice of pro-environmental behaviour. The rest of this article is structured as follows: Section 2 presents an overview of the literature and theoretical foundation in relation to the links between environmental commitment, environmental consciousness, and green lifestyle, green self-efficacy, and pro-environmental behaviour, and sets out the hypotheses to be tested. Section 3 outlines the research method. Section 4 present the results, and Section 5 details the analyses and discussion of the results, implications, limitations, and future research directions. Section 6 closes the article with a conclusion.

## 2. Literature Review

### 2.1. Pro-Environmental Behaviour

Pro-environmental behaviour includes recycling (e.g., reusing paper, plastic, glass, containers), conserving water (e.g., limiting the use of water when taking a shower or washing hands), saving electricity (e.g., turning off lights when not needed), reusing (e.g., disposable cups), using public transportations or riding bikes or even walking, properly disposing of non-recyclable waste, using less paper when printing (e.g., double-sided printing), and buying and/or consuming green products [48].

Pro-environmental behaviour may be influenced by various aspects [23], such as socio-demographics—gender, age or residence, political perspective, values, and beliefs about life [49]. Additionally, as Vicente-Molina et al. [23] opined, this behaviour can be changed by public-sphere behaviour (e.g., public policies). Pro-environmental behaviour can be directly affected by the private and public spheres, with examples including consumption of green products, use of public transportation, and recycling. Individuals' intention to practice responsible environmental behaviour is not only influenced by personal beliefs but by others' behaviours and actions. In the university setting, as Vicente-Molina et al. [23] presumed, pro-environmental behaviour among students can be promoted by university' plans and actions, such as providing disposable containers or offering environmental-related subjects.

### 2.2. Environmental Commitment

According to Rahman and Reynolds [39], individuals may feel committed to the environment when they feel psychologically connected to and familiar with nature. Two theories related to environmental commitment are interdependence theory, which explains the relationships between two people, and commitment model, which examines how commitment comes about. Based on the two theories, it can be argued that individuals are willing to be committed to a person,

a place, or the environment when they have to rely solely on any of these for meeting their needs. Environmental intentions, intended behaviours, and preferences can be predicted by biospheric value orientation [11]. Biospheric values are values that show concern for the environment. High degree of environmental commitment tends to correlate with strong biospheric values [40].

Han and Hyun [11] and Rahman and Reynolds [39] speculated that environmental commitment can determine individuals' readiness to do what is necessary for the sake of the environment. Environmental commitment of individuals has been shown to be a good determinant of their past green behaviours and their intended green behaviours. For instance, individuals tend to positively focus on the intention to consume green products and practice pro-environmental buying orientation. Liu and Lin [50] posits that individuals' environmental behaviour and commitment towards the environment would be affected by mental model (e.g., mental image).

## 2.3. Environmental Consciousness

Environmental consciousness is acknowledged as a psychological aspect which determines individuals' tendency towards pro-environmental behaviour, and it consists of multidimensional constructs that have effects on individuals' attitudes, behaviours, knowledge, actions, and intentions [15]. Environmental consciousness (also referred to as environmental awareness) lies in cognitive factors, and includes environmental value, environmental concern, environmental knowledge, and self-efficacy that emphasize individuals' understanding that influences their particular behaviours and indicates the degree of their awareness of the negative impacts of refusing to engage in pro-environmental behaviour [11,51].

In Martinez Garcia de Leaniz et al. [52], the concept of environmental consciousness was investigated in the context of hotel guests, where they were measured in terms of their concern on environmental issues and actions taken to help resolve the issues, such as by staying at green hotels. Ari and Yilmaz [4] construed that environmental consciousness leads individuals to engage in environmentally friendly behaviours to help minimize the environmental contamination, including consuming recyclable and eco-friendly products, participating in environmental activities and programs, and purchasing products by companies that advocate environmental sustainability. Apart from that, environmental consciousness is interrelated to environmental education that motivates individuals to practice and engage in pro-environmental behaviour [16].

## 2.4. Green Lifestyle

Green lifestyle encompasses various aspects of individuals' intention to preserve the environment. Axsen et al. [1] defined green lifestyle, or pro-environmental lifestyle, as an individual's engagement (e.g., a way of living) in various types of pro-environmental tasks and activities. In the context of green consumer behaviour research, green consumer lifestyle can be predicted by their green purchase decision. Kumar and Ghodeswar [53] and Lai and Cheng [54] posit that green consumerism or pro-environmental purchase behaviour is part of environmentally-concerned behaviour of individuals. Green lifestyle also refers to pro-social behaviour or pro-environmental behaviour that shows responsibility towards nature [55]. This is shown, for example, through the use of green products, which are environmentally and ecologically friendly products that integrate recyclable materials or contents that serve to protect the environment [12].

Green or environmentally-oriented individuals tend to choose eco-friendly products because they are concerned about their health and the environment, and before purchasing they always engage in product evaluations, including finding out how far the products can be recycled and disposed of and how they are produced [12]. In addition, Sony and Ferguson [56] argue that pro-environmental consumers prefer recycled products to new ones that are not environmentally friendly.

## 2.5. Green Self-Efficacy

Green self-efficacy is the belief individuals have that they can take actions to improve environmental quality [13,57] regarded efficacy as a valuable component in human competence development because it can adjust individuals' sensations, thinking formations, actions, and motivations. Individuals' perception of efficacy influences their intention to engage in pro-environmental behaviour when there exist elements of fear (negative emotions associated with physiological arousal) and/or threat (perception of the received external message which individuals think they are exposed to adverse effects or conditions) [58].

According to Tabernero and Hernandez [58], self-efficacy judgments would affect individuals' targeted goals and their affective reactions towards the performance accomplished in different settings. The authors further added that self-efficacy is capable of assisting individuals to concentrate on their attention and reduce nature degradation; influence the degree of the perceived difficulty of targeted goals and commitment level to achieve those goals; and point out more productive strategies and efficiently facilitate resources towards the targeted goals. Self-efficacy has been shown to positively influence pro-environmental behaviour by affecting individuals' internal motivation and further adjusting their pro-environmental behaviour simultaneously [59]. The behaviour continues to grow over time, and the individuals learn how to perform particular activities in a pro-environmental manner [14].

## 2.6. Theoretical Foundation

In previous studies, there were various theories used to predict pro-environmental behaviour [13]. Values theories, including egoistic, altruistic and biospheric value orientation, serve as guiding principles in individuals' life that can affect their attitudes, beliefs, and behaviours that lead to the pro-environmental behaviour [39]. Furthermore, Rahman and Reynolds [39] stressed that biospheric value orientation (given priority to take care of the environment) is more significant in terms of forcing individuals' behavioural intention because of its relevance in the process of developing pro-environmental behaviour.

Theory of planned behaviour (TPB) encompassing three motivating factors—attitude, subjective norm, and perceived behavioural control—was proposed by Ajzen [59] as a widely used theoretical foundation to determine intended behaviours. The theory was set forth from the initial theory of reasoned action (TRA). It was mainly based on the underlying assumption that the reason of consciousness evolved in the construction of individuals' intention to execute particular behaviours, and those behaviours are partially under individuals' control [59]. The TPB model hypothesizes that environmental beliefs develop the attitudes towards intended behaviours, and the behavioural intentions lead to the engagement of the pro-environmental behaviour [39].

For example, the TPB model was used in a study by Halder et al. [60] to determine the intention to use bioenergy among high school students in three high schools in India. The authors argued that the hypothesized TPB model was able to explain the significant variances in pro-environmental behaviour and pro-environmental intentions in a cross-cultural setting. Chen [61] manipulated and extended TPB model with moral obligation to examine individuals' intention to conserve energy, reduce carbon emission, and help mitigate climate change in the context of the pro-environmental behaviour.

Self-determination theory was used in a study to determine the role of employees' own motivation in influencing pro-environmental behaviour when external motivations such as rewards for performing pro-environmental behaviour were removed [14]. In addition, the norm-activation model was used in the investigation of pro-environmental behaviour and environmental actions to explain individuals' altruistic behaviours that focused on moral obligations [57].

*2.7. Hypotheses Development*

2.7.1. Relationship between Environmental Commitment and Pro-Environmental Behaviour

Rahman and Reynolds [39] found that consumers' environmental commitment in the form of willingness to sacrifice for the environment, when included as mediator in the relationship between biospheric values orientation and green hotel-specific behavioural intentions, did influence green hotel-specific behavioural intentions. Han and Hyun [11] found that willingness to sacrifice for the environment and nature connectedness were significantly relevant as predictors of visitor' intentions to visit museums in South Korea, and the high level of visitors' willingness led to pro-environmental intentions and behaviours.

Liu and Lin [50] revealed that undergraduate students in Taiwan with higher scores of environment mental model exhibited higher emotional connection and commitment towards the environment, suggesting that it is a vital factor in fostering pro-environmental behaviour. Greater commitments exhibited by individuals towards the environment likely lead to their engaging in pro-environmental behaviour. The argument is supported by Hergesell [62], where environmentally committed travelers put greater consideration on environmental consequences when choosing a mode of transport during trips (e.g., using train more and car less) regardless of comfort and travelling time.

Davis, Le and Coy [62] stated that individuals with a high degree of satisfaction and investment towards the environment are more likely to have a high degree of environmental commitment, which in turn drives them to engage in pro-environmental behaviour. Terrier and Marfaing [63] suggests that environmental commitment tends to strengthen individuals' perception of themselves, thereby motivating them to be environmentally friendly individuals. The argument was supported in their study in that initial commitment portrayed by hotel guests to supporting hotels' pro-environmental initiatives (i.e., reuse of bath towels) led them to engage in pro-environmental behaviour on their free will. Hence, it can be seen that individuals' environmental commitment may be an important predictor of students' pro-environmental behaviour. Therefore, the following hypothesis was posited:

**Hypothesis 1 (H1).** *Environmental commitment is positively related to pro-environmental behaviour.*

2.7.2. Relationship between Environmental Consciousness and Pro-Environmental Behaviour

Ari and Yilmaz [4] found that environmental awareness of middle school students in Turkey influenced their pro-environmental attitudes, which in turn positively affected their pro-environmental purchasing behaviour. Further, Mishal et al. [15] demonstrated that environmental consciousness of consumers in India significantly influenced their green purchase attitude, thereby influencing their green behaviour. Martinez Garcia de Leaniz et al. [52] proved that hotel guests with high degree of environmental consciousness tended to stay in green hotels and were willing to pay premium prices to stay in these environmentally certified hotels. Zareie and Navimipour [21] revealed that electronic environmental knowledge consisting of environmental attitudes, environmental awareness, environmental values, public information, environmental skills, and environmental responsibility positively influenced environmental behaviours of undergraduates and postgraduates in Iran. Khare [8] found that consumers in India with awareness of their environmental responsibilities and concerned for the conservation of the environmental resources were interested in engaging in green buying behaviour, such as purchasing green and recycled products. Blok et al. [5] stated that individuals with greater environmental knowledge and ability to grasp environmental issues showed more interest in pro-environmental behaviour. Environmental consciousness appears to be one of the critical factors that may predict students' pro-environmental behaviour. Hence, the following hypothesis was proposed:

**Hypothesis 2 (H2).** *Environmental consciousness is positively related to pro-environmental behaviour.*

### 2.7.3. Relationship between Green Lifestyle and Pro-Environmental Behaviour

Lai and Cheng [54] argued that pro-environmental attitudes of undergraduates in Hong Kong and their willingness to pay for eco-friendly products led them to practice green purchase behaviour or pro-environmental behaviour, as a way of displaying responsibility towards the environment. Kumar and Ghodeswar [53] revealed that Indian consumers actively using environmentally friendly products as part of their green lifestyle felt encouraged to practice pro-environmental behaviour.

Consumers who adopt green lifestyle are more environmentally conscious and, therefore, have a higher degree of sensitiveness towards their own behaviours, and this urges them to adopt pro-environmental behaviour to help alleviate environmental issues. Mohd Suki [12] and Mohd Suki and Mohd Suki [64] found that Malaysian university students' green lifestyle influenced their satisfaction and loyalty towards using green, eco-friendly, recyclable products, which are part of pro-environmental behaviour. Axsen et al. [1] postulates that individuals who practice pro-environmental or green lifestyle, including recycling, using pro-environmental technology (e.g., driving a hybrid vehicle), and consuming organic foods, are likely to engage in pro-environmental behaviour when trying out new approaches and products. It can be seen that green lifestyle can be used as one of the determinants of pro-environmental behaviour. In addition, Ting et al. [18] argued that younger generations have higher tendency to engage in individual behavioural change and modification of their lifestyle to adjust to different condition. Therefore, it is worthwhile to examine whether students' green lifestyle can be a predictor of pro-environmental behaviour. Hence, the study proposed the following hypothesis:

**Hypothesis 3 (H3).** *Green lifestyle is positively related to and pro-environmental behaviour.*

### 2.7.4. Relationship between Green Self-Efficacy and Pro-Environmental Behaviour

Self-efficacy is one of the cognitive factors that shapes pro-environmental behaviour [11]. Huang [13] found that individuals in Taiwan with higher level of environmental belief and self-efficacy tended to engage in all types of pro-environmental behaviour, including promotional, accommodating, and proactive behaviour. Kim et al. [14] found that employees who have high self-efficacy in the hospitality industry in Seoul were more inclined of their own volition to adopt eco-friendly goals at the workplace as part of their action to help reduce environmental degradation. Tabernero and Hernandez [58] argues that individuals with high degree of self-efficacy tend to engage more in pro-environmental behaviour and behave environmentally than those with low degree of self-efficacy.

According to Chen [57], undergraduates in Taiwan who were grouped under high-fear appeal condition had higher moral obligation to alleviate environmental degradation, thereby increasing their intention to engage in pro-environmental behaviour. As argued by Abraham et al. [65], self-efficacy can be a reliable predictor of pro-environmental behaviour. They found that the greater the level of undergraduates' self-efficacy, the more persistent and determined they were to counter the environmental problems, which enhanced their intention to engage in pro-environmental behaviour. Jugert et al. [66] provided empirical evidence that manipulation of collective efficacy influenced perceived collective and self-efficacy, which in turn affected university students' pro-environmental intentions and pro-environmental behaviour. It can be seen that self-efficacy may serve as a reliable predictor of pro-environmental behaviour among students. Therefore, the following hypothesis was proposed:

**Hypothesis 4 (H4).** *Green self-efficacy is positively related to pro-environmental behaviour.*

The research framework is presented in Figure 1.

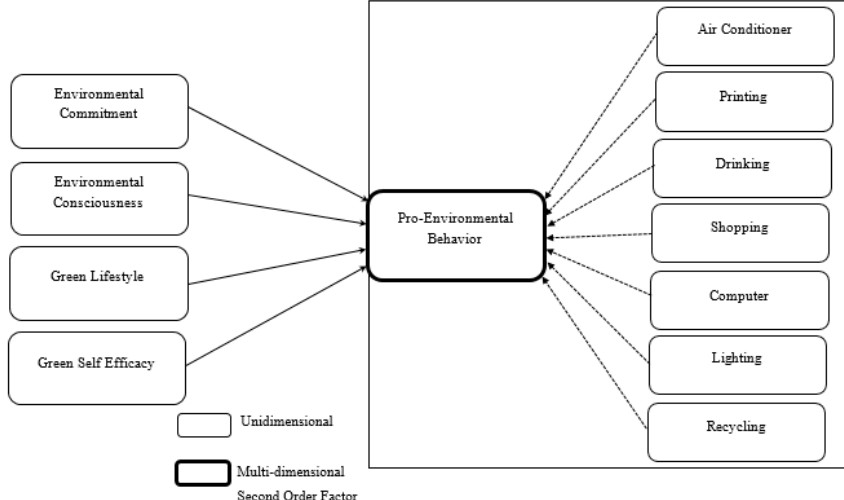

**Figure 1.** Research Framework.

## 3. Materials and Methods

### 3.1. Sample and Procedure

Using correlational design, the present study aimed to examine the relationships between environmental commitment, environmental consciousness, green lifestyle, green self-efficacy, and pro-environmental behaviour. To test the research model, a survey instrument in the form of questionnaire was designed and measurement scales were developed. With the help of four academics, the scales were pre-tested to check the content validity. The final improved version of the questionnaire was then used to test the proposed hypotheses. The sample for this study was selected using convenience sampling method. It consisted of 85 students of a training centre known for its technical timber, design, and building architecture education. The students were given the survey questionnaires during their involvement in a green awareness talk on 20 November 2019. The questionnaires were accompanied by cover letters that included brief explanation of the research and assurance of the students' anonymity. Before they answered the questionnaire, they were informed about the research objectives, data collection methods, and the right to withdraw from the study and informed that their participation is voluntary. Out of the 85 students, 13 did not return the survey questionnaire. A total of 72 effective responses were received, indicating a valid response rate of 85 percent.

### 3.2. Measurements

Pro-environmental behaviour was measured by 27 items that covered the use of air-conditioning, printing, copying, sustainable drinking, sustainable shopping, use of computer, lighting, and recycling. These measurement items were adapted from Blok et al. [5]. Responses for each item were based on a 6-point Likert-type scale consisting of (0) not related, (1) never, (2) rarely, (3) sometimes, (4) often, and (5) always. Environmental commitment was measured by 8 items adapted from Raineri and Paille [67]. Responses for each item were based on a 5-point Likert-type scale ranging from (1) strongly disagree to (5) strongly agree. Green lifestyle was measured by 7 items adapted from Pickett-Baker and Ozaki [68] and Sony and Ferguson [56]. Responses for each item were based on a 5-point Likert-type scale ranging from (1) never to (5) always. Environmental consciousness was measured by 4 items, which were adapted from previous studies [69–72]. Responses for each item were based on a 5-point Likert-type scale ranging from (1) strongly disagree to (5) strongly agree. The last factor, green self-efficacy, was measured by 6 items adapted from Chen et al. [73]. Responses for each item

were based on a 5-point Likert-type scale ranging from (1) strongly disagree to (5) strongly agree. Measurement items are provided in the Appendix A.

*3.3. Data Analysis*

To test the model, second-generation statistical software SmartPLS 3.2.9 was used [74]. As suggested by Hair et al. [75], the measurement model was assessed by conducting a Confirmatory Composite Analysis (CCA); then the structural model was analyzed using a 5000 resample bootstrapping method. One of the reasons for using SmartPLS was because pro-environmental behaviour model in this study was based on a Type II (Reflective-Formative) measurement.

## 4. Results

*4.1. Demographic Profiles of Respondents*

Of the 72 respondents, 69.4 percent were male, and 30.6 percent were female. Most of them (72.22%) were between the ages of 16 and 19, a few (25.0%) were between the ages of 20 and 23, and the rest (2.78%) were above the age of 24. The majority of the students (97.2%) were Malay, while only 1.0 percent were Chinese, and another 1.0 percent Indian. For educational background, 37.5 percent of the students had a Lower Certificate of Education (PMR). Another 36.1 percent had a Malaysian Certificate of Education (SPM). A few (16.7%) possessed a Diploma, while 9.7 percent had other educational certificates. As for marital status, a total of 95.8 percent indicated they were single, 2.8 percent divorced, and 1.4 percent married.

*4.2. Measurement Model Assessment*

Validity and reliability were first assessed by looking at the loadings, average variance extracted (AVE) and the composite reliability (CR) [75,76]. As shown in Table 1, all the AVE were greater than 0.5, and the CR were greater than 0.7. However, some loadings were lower than 0.7. Since the AVE and the CR all met the required cut-off values, it was concluded that the measurement had convergent validity and reliability [77].

**Table 1.** Mean, Standard Deviation, Average Variance Extracted and Composite Reliability.

| Construct | CR | AVE | Mean | SD |
|---|---|---|---|---|
| Air Conditioner | 0.875 | 0.778 | 3.793 | 1.257 |
| Computer | 0.942 | 0.890 | 4.227 | 1.210 |
| Drink | 0.768 | 0.528 | 3.007 | 0.959 |
| Environmental Commitment | 0.892 | 0.509 | 3.874 | 0.670 |
| Environmental Consciousness | 0.778 | 0.541 | 3.703 | 0.614 |
| Green Lifestyle | 0.862 | 0.517 | 3.215 | 0.853 |
| Green Self Efficacy | 0.909 | 0.626 | 3.462 | 0.611 |
| Lighting | 0.880 | 0.710 | 4.290 | 0.896 |
| Print | 0.838 | 0.633 | 3.423 | 0.996 |
| Recycle | 0.894 | 0.739 | 3.564 | 0.978 |
| Shopping | 0.684 | 0.543 | 4.106 | 1.160 |

Since pro-environmental behaviour was modelled as Type II second-order measurement, the quality of the formative measurement was first assessed by looking at the weights, *t*-values and the variance inflation factor (VIF). As Hair et al. [76] suggested, VIF not exceeding 5 indicates multicollinearity; based on Bollen [78], all the dimensions were retained, as theoretically all these form the pro-environmental behaviour. Table 2 displays assessment for formative measurement.

**Table 2.** Assessment for Formative Measurement.

| Dimensions | Weights | Std. Error | *t*-Value | *p*-Value | VIF |
|---|---|---|---|---|---|
| Air Conditioner | 0.532 | 0.127 | 4.198 | 0.000 | 1.325 |
| Computer | 0.145 | 0.147 | 0.987 | 0.162 | 1.391 |
| Drink | 0.057 | 0.14 | 0.408 | 0.342 | 1.387 |
| Lighting | 0.044 | 0.168 | 0.263 | 0.396 | 1.734 |
| Print | 0.500 | 0.156 | 3.210 | 0.001 | 1.365 |
| Recycle | 0.066 | 0.201 | 0.329 | 0.371 | 1.165 |
| Shopping | 0.285 | 0.161 | 1.773 | 0.038 | 1.242 |

Next, discriminant validity was assessed by looking at the HTMT ratios. As Franke and Sarstedt [79] suggested, discriminant validity has no issue if the ratios are lower than HTMT0.85 criterion. As shown in Table 3, all the HTMT ratios were lower than the cut off value of 0.85, thereby confirming the measurements were distinct.

**Table 3.** Discriminant Validity (HTMT Criterion).

| Constructs | 1 | 2 | 3 | 4 | 5 |
|---|---|---|---|---|---|
| 1. Environmental Commitment | | | | | |
| 2. Environmental Consciousness | 0.832 | | | | |
| 3. Green Lifestyle | 0.210 | 0.195 | | | |
| 4. Green Self Efficacy | 0.703 | 0.616 | 0.231 | | |
| 5. Pro-Environmental Behaviour | 0.667 | 0.280 | 0.247 | 0.557 | |

*4.3. Structural Model Assessment*

For hypothesis testing, a bias-corrected bootstrapping with 5000 resample was run to generate the t-values. To test the in-sample predictive accuracy, R2 values are observed. The R2 value was 0.511 (Q2 = 0.439), which indicated that the four independent variables together could explain 51.1% of the variance in pro-environmental behaviour with an acceptable predictive relevance of 0.439, which was greater than 0 [80].

The four developed hypotheses posited each factor to have a positive effect on pro-environmental behaviour. Based on Table 4, environmental commitment ($\beta = 0.635$, $p < 0.01$), environmental consciousness ($\beta = 0.285$, $p < 0.01$), green lifestyle ($\beta = 0.155$, $p < 0.01$) and green self-efficacy ($\beta = 0.245$, $p < 0.01$) all positively affected pro-environmental behaviour. Thus, H1, H2, H3, and H4 were all supported.

**Table 4.** Hypotheses testing.

| Hypotheses | Relationship | Std Beta | Std Error | *t*-Value | *p*-Value | BCI LL | BCI UL | $f^2$ |
|---|---|---|---|---|---|---|---|---|
| H1 | Env Commitment→ Pro-Env Behaviour | 0.635 | 0.136 | 4.675 | $p < 0.001$ | 0.421 | 0.857 | 0.400 |
| H2 | Env Consciousness → Pro-Env Behaviour | 0.285 | 0.139 | 2.047 | 0.020 | 0.096 | 0.554 | 0.101 |
| H3 | Green Lifestyle → Pro-Env Behaviour | 0.155 | 0.076 | 2.039 | 0.036 | 0.057 | 0.307 | 0.047 |
| H4 | Green Self Efficacy → Pro-Env Behaviour | 0.245 | 0.103 | 2.377 | 0.009 | 0.080 | 0.415 | 0.072 |

## 5. Discussion

The originality of this study is in putting together the relationships between environmental commitment, environmental consciousness, green lifestyle, green self-efficacy, and pro-environmental

behaviour. This section first discusses the study's main findings, then its theoretical and practical implications, limitations, and future directions.

### 5.1. Main Findings

The findings indicated that students' environmental commitment, environmental consciousness, green lifestyle, and green self-efficacy had significant positive effects on their pro-environmental behaviour. Students' environmental commitment influenced their pro-environmental behaviour most significantly. This is consistent with previous findings which have suggested that individuals' willingness to sacrifice and be environmentally committed towards the environment shape their pro-environmental behaviour [11,39]. The students were able to comprehend their responsibility towards nature, and this in turn stimulated and extended their feelings of commitment to amending the environmental issues through their engaging in the pro-environmental behaviour [50].

Furthermore, students' commitments towards the environment were high. For example, they rode bicycles within campus, regardless of the other factors (time and comfort) so as to portray their concern on the consequences that might arise from their actions, such as air pollution [81]. Such commitment stemmed from their high sense of satisfaction when investing towards nature to ensure that the goal of environmental preservation could be achieved [62]. In addition, their full commitments towards supporting environmental sustainability were aided by their feeling of attachment to the training center and its surrounding.

The findings indicated that students' environmental consciousness played a crucial role in influencing their pro-environmental behaviour. Environmental consciousness is the process of connecting individuals to their surrounding and preservation of nature. This means that students' awareness enabled them to develop positive environmental behaviours and propelled them to engage in pro-environmental conducts [21,52]. Students' exposure to environmental issues through the activities in college, knowledge, lifelong learning, or peer pressure enabled them to grasp these current environmental issues and made them consciously alert with the associated impacts [5]. For instance, students supported the initiatives by disposing of materials into the recycling containers and using disposable drink cups provided in the cafeteria. Besides, the results are also in line with those in previous studies, whereby individuals with a high level of environmental consciousness intended to purchase green products (green buying behaviour), consume eco-friendly products, and participate in environmental activities to support environmental sustainability missions [4,8,15,82]. Therefore, it can be confirmed that students are now aware of and conscious about environmental issues, through environmental activities, lifelong learning, experiences, and knowledge that help them foresee risks associated with environmental problems originated from human behaviours.

The hypothesized model proposed that students' green lifestyles are positively related to their pro-environmental behaviour. The findings indicated that students' green lifestyle, such as utilizing recyclable products and buying green products, did influence their pro-environmental behaviour. This is consistent with previous studies by Kumar and Ghodeswar [53] and Lai and Cheng [54], in which pro-environmental attitudes and willingness to pay and consume environmentally friendly products were the influential leading factors that shaped pro-environmental behaviour. Students adopted green lifestyle in light of the recognition of their responsibility to protect the environment [66]. In addition, Axsen et al. [1] reported that individuals tended to adopt green or healthier lifestyle on account of having trendier lifestyle evoked by social groups and the desire to try new things. Subsequently, green lifestyle adoption led them to engage in pro-environmental behaviour including recycling, using pro-environmental technology, and preferring organic foods. College students are from the younger generations that are exposed to immediate changes [18]. For this reason, students tend to follow a trendy lifestyle, such as a green lifestyle, by their own choice, and modify their lifestyle according to the current stream. Further, students are more inclined to practice green lifestyle when they have pro-environmental attitude, and this attitude motivates them to purchase green products that appeal to their interest and satisfaction. When this happens, students have engaged in pro-environmental

behaviour [12]. Furthermore, it does not matter if the students practice green lifestyle in other settings (individually or collectively) or not, but rather that a green lifestyle encourages students to display their commitments towards environmental problems.

The findings indicated that green self-efficacy positively influenced pro-environmental behaviour of students. The reasoning lies in the studies by Chen [57] and Jugert et al. [67], where it was demonstrated that manipulation of collective efficacy of university students influenced their self-efficacy towards climate change. The orientation of collective efficacy can affect individuals' self-efficacy to act pro-environmentally in daily decisions, although they are not affected by the changes in climate. Furthermore, individuals' self-efficacy is growing over time, which enables individuals to discover how to execute tasks in a pro-environmental way and adopt pro-environmental behaviour [11,13,57]. This shows that students were willing to perform pro-environmental behaviour according to their capability to attain environmental goals. For example, students knew where to dispose of recyclable materials (e.g., bottles, plastics, and papers), ensured that electricity was turned off after class, or saved paper even though they were not told to do so. Further, the results of the present study supported the argument by Abraham et al. [65] and Kim et al. [14], which stated that a high level of self-efficacy impelled individuals to become more consistent in countering environmental issues, thereby increasing their pro-environmental behaviour.

### 5.2. Theoretical Implications

The present study contributes to the literature and to the understanding of the connection between students' environmental commitment, environmental consciousness, green lifestyle, and green self-efficacy and their engaging in pro-environmental behaviour. The results showed that all the variables had positive direct effects on pro-environmental behaviour. The results can indirectly be used to extend the literature on the TPB model although the hypotheses developed did not solely depend on the model. This is in line with Chen [61] which posits that the TPB model plays a major role in describing and explaining social psychological constructs over time that correlate behavioural intention and actual behaviours. The results lend support to the arguments of the TPB model in that students' environmental beliefs and values shape their attitudes that subsequently lead to intended environmental behaviours that further drive them to practice pro-environmental behaviour.

As also shown in previous studies by Liu and Lin [50] and Han and Hyun [11], students' environmental commitment is an essential construct for comprehending determinants of pro-environmental behaviour. This is important as it has been shown that when individuals have low level of commitment towards the environment, they tend to ignore the adverse circumstances from their actions. In addition, it is important to investigate the relationship between environmental consciousness and pro-environmental behaviour. This sheds lights on the significance of environmental education and knowledge for enhancing environmental consciousness to foster individuals' pro-environmental behaviour [16] that is crucial to environmental sustainability [51]. Hence, this can pave the way for educational institutions to extensively plan and design environmental courses and curriculums to produce more environmentally responsible students. Furthermore, the findings extend our understanding of the influence of green lifestyle on pro-environmental behaviour. Green lifestyle is rather actively utilized as a trigger for pro-environmental behaviour when developing a conceptual model, for the reason that individuals favoring green consumerism are likely to make less contamination as possible, such as by purchasing green products [54]. Likewise, green self-efficacy is related to pro-environmental behaviour. This study provides justification that individuals with higher perceived self-efficacy are more likely to act in a pro-environmental way and engage in pro-environmental behaviour [14]. Consequently, once individuals have strong self-efficacy, they will exert considerable commitments and efforts to protect the nature.

This study contributes to the existing environmental sustainability literature by focusing on training center students in Malaysia. The intensification of environmental awareness and education towards the conservation of nature has created heightened interest in the present times in pro-environmental

behaviour [22]. This study focused on students because they are the main group that can protect the future of environmental sustainability both in colleges and public settings for long-term. It is also worthwhile to increase students' understanding of what responsible students need to do towards maintaining sustainable relationships.

### 5.3. Practical Implications

The results have several practical implications. Although this study took place at one of the training centers, it has pertinent contributions towards educational institutions in general, including schools, foundations, and universities. The justification is based on the premise that environment-based education has become the preferred choice for educational institutions in accomplishing environmental sustainability through the imparting of valuable knowledge, abilities, and experiences to the students [50].

It may be imperative for an educational institution to introduce environmentally related subjects and activities to all students that mainly focus on emotional orientation instead of cognitive orientation. Students' participation in these subjects and activities will boost their intention to behave environmentally and become sensitive to environmental issues (related with feelings and emotions) as well as enhance their understanding about the environmental system. These aspects potentially can shape and foster students' environmental behaviours since emotion is of the main predictors of environmental behaviours [6]. For example, Vicente-Molina [23] argues that environmental training projects are beneficial for increasing positive environmental viewpoints, behaviours, and attitudes of the groups.

This study has contributed towards enhancing top management' understanding that sufficient and continuous assessment processes are essential to evaluate and monitor students' impacts on the environment to achieve efficiency and effectiveness in the attainment of environmental sustainability goals [83,84]. This can be done through implementation of students' environmental know-how assessment that can be evaluated by the lecturers. These assessments can be performed twice per semester at the beginning and the end to provide insights for top management about current changes in students' attitudes and behaviours toward environmental sustainability. Monitoring provides feedback on whether implementation taken is pointing in the right direction. The assessment of the students is vital to produce capable students that can help mitigate environmental issues and that can take actions required to reduce adverse impacts on the surroundings.

Another important implication is this study can benefit governments and private agencies, as they can gain necessary information related to students, which can help their initiatives and strategies to sustain the environment. A study by Huang [13] showed that individuals were keen to access and gather information about global warming through the use of media such as the internet, newspapers, and television, which positively affected their pro-environmental behaviour. For that reason, the same medium including environmental campaigns, programs, advertisements, and news provided by the government and private agencies can help escalate students' awareness. For instance, information can be about current issues, ways to preserve the environment, green lifestyle. It can also highlight possible adverse outcomes due to lack of concern for environmental preservation. Social media can help engage the younger generations towards the environment.

### 5.4. Limitations and Future Research Directions

There were several limitations of the study. First, the samples were taken only from a timber training centre in the country. This implies that they might not reflect different situations that may exist in different countries, so the results cannot be generalized to these countries. Future studies should be improved from the aspect of sample to include larger size and different target groups with different educational levels and job categories to enable better understanding of pro-environmental behaviour. Second, data were gathered cross-sectionally. Longitudinal studies are needed. Third, data were collected using a questionnaire surveys, focus group might be useful in complementing

questionnaire designs and supplement results discussion in future research. Fourth, self-reported data may contain biases compared to behavioural measures. Fifth, although the variables selected for the framework can be considered adequate for pro-environmental study in training centre, it is believed that other variables can also contribute to this framework. The possible association amongst the triple bottom line of sustainability can further be explored, as well as the inclusion of constructs such as environmental attitude, environmental knowledge and concern, environmental awareness, green mindfulness, green climate, and organizational citizenship behaviour towards the environment. Finally, more research is needed to clarify these results using multiple methods (i.e., moderation models to test the potential influence of gender [85] in the link between independent variables and pro-environmental behaviour). It can be argued that different genders have different attitudes and behaviours that affect pro-environmental behaviour [50]. The authors added that females across cultures tend to have a set of compelling attitudes, including helpful, sympathetic, nurturing, and expressive while males are more competitive and self-reliant. Meyer [19] also noted that, in the broad population, demographic factor like gender is an important factor in the context of pro-environmental behaviour because male and female have significant differences in their perceptions towards pro-environmental behaviour. Future studies can take into account the gender factor. It will be interesting to observe the influence of gender in pro-environmental behaviour in terms of which gender has stronger effects and gives better responses to environment-based practices [23]. The methods and techniques of analysis used might, as suggested by some scholars, influence the results, notably if there are mediating effects as there seem to be in the case of pro-environmental behaviour. Therefore, using techniques that take these aspects into accounts when analysing pro-environmental behaviour is recommended. Otherwise the results might point to misleading conclusions. Each of these limitations can be ameliorated by future research.

## 6. Conclusions

In recent decade, the growth of interest in green campus is observed. The current research adds to the broader literature on pro-environmental behaviour. This study investigated whether students' pro-environmental behaviour stemmed from their initiatives or was enforced through the training centre' rules and regulations. We justify the framework and bridge prior work examining the relationship between environmental commitment, environmental consciousness, green lifestyle, green self-efficacy, and pro-environmental behaviour into a cohesive model. Our findings suggest that students' environmental commitment, environmental consciousness, green lifestyle and green self-efficacy plays a vital role in influencing the pro-environmental behaviour of the students. One of the startling findings in this study was that students had strong environmental understanding and intention to protect the nature. This can be attributed to the fact that students have been more aware of and more alert with environment-related activities either from their self-awareness or due to the holistic approach of the curriculum and direction of the training centre itself. Pro-environmental behaviour has been widely acknowledged in universities that seek to be part of 'green universities', whereby the role of the students is pivotal to achieving environmental sustainability goals. Above all, environmental-based knowledge and education are crucial for the younger generation for the preservation of nature and future continuity.

The present study made a number of important contributions to our understanding of pro-environmental behaviour. First, it is one of the few studies to examine the relationship between environmental commitment, environmental consciousness, green lifestyle, green self-efficacy, and pro-environmental behaviour in a single study. Second, the participants in our study were timber training centre students, arguable an important population because eco-friendly habits may be established early in their study life in the training centre and because timber training centre students are exposed to environmental education related to the amount of waste generated from wood as part of their curriculum. In this regard, finally our discussion above shows how the results of the present

study can provide useful guidance regarding the kinds of factors to be considered in designing an effective behaviour change intervention.

In conclusion, training centre will be able to act as learning institutions by educating and training the young generation in technical background to secure the future of the global society. Academic freedom and autonomy enable training centre to perform a central role in developing green mindset students as well as environmental learning systems towards sustainable development.

**Author Contributions:** Conceptualization, M.Y.Y., T.R. and N.A.N.S.A.; methodology, J.S., S.M., and T.R.; software, J.S.; validation, T.R., M.Y.Y. and J.S.; formal analysis, M.Y.Y., J.S. and N.A.N.S.A.; investigation, S.M., R.A.R.; resources, M.Y.Y., and N.A.N.S.A.; data curation, M.Y.Y. and Z.M.; writing—original draft preparation, M.Y.Y, Z.M. and N.A.N.S.A.; writing—review and editing, J.S., A.A., R.A.R., F.D.M. and M.M.; visualization, S.M.; supervision, M.Y.Y., Z.M.,S.M., and T.R.; project administration, Z.M.; funding acquisition, M.Y.Y. All authors have read and agreed to the published version of the manuscript.

**Funding:** This work was supported by the Knowledge and Technology Assimilation Grant Scheme from Universiti Malaysia Terengganu (Vot. No. 58902).

**Acknowledgments:** We would like to thank for Universiti Malaysia Terengganu that has supported this research and publication under Knowledge and Technology Assimilation Grant (KTAG) Scheme.

**Conflicts of Interest:** The authors declare no conflict of interest. Mohd Yusoff Yusliza has received research grant from Universiti Malaysia Terengganu. Zikri Muhammad and Safiek Mokhlis is a member of the same research grant from Mohd Yusoff Yusliza.

## Appendix A. Pro-Environmental Behaviour

### *Appendix A.1. Air-Conditioning*

Please indicate how you use the air-conditioning in the campus.

1. I check whether the temperature is set correctly in the classroom. The ideal setting is 24 °C or more.
2. I make sure that air-conditioning is off after the class.
3. I make sure that the temperature is increased outside learning hours.
4. I make sure that the temperature is increased in unused room.
5. I make sure that air-conditioning is off in unused room.

### *Appendix A.2. Printing*

How often do you do following activities related to printing and copying in the campus?

1. I try to get as much as possible to print on both pages one A4 sheet.
2. I try to get as much as possible to print two pages on one A4 sheet.
3. I copy double-sided on one A4 sheet.
4. I try to get as much as possible on one sheet by using narrow margins on one A4 sheet.

### *Appendix A.3. Drinking*

To what extent do the following statements suit you when you are in campus?

1. I use a mug for drinking water.
2. I wash the mug in a sustainable way by using cold water, no use of washing-up liquids.
3. I take a new plastic/carton cup each time I have a drink.
4. I use stainless steel straw when drinking.
5. I try to reduce the use of straw when drinking.

### *Appendix A.4. Sustainable Shopping*

To what extent do following statements suit you when you are in the campus?

1. I choose bio food when if it is offered in a cafeteria in the campus.
2. I bring my own shopping/plastic bag when I shop in the campus.
3. When I purchase goods or services, I pay attention to sustainability.

*Appendix A.5. Computer Use*

To what extent do following statements suit you when you are in the campus?

1. I switch off my computer/notebook when I leave the class for a considerable period.
2. I switch off my computer/notebook when I go home/hostel.

*Appendix A.6. Light Use*

To what extent do following statements suit you when you are in the campus?

1. I switch on the lights when I come to the class and switch them off after my class.
2. I switch off the lights when I leave my class for a considerable period of time if there is no one else in the class.
3. I switch off the lights when there is no one else in the class.

*Appendix A.7. Recycling*

To what extent do you recycle the following products when you are in the campus?

1. Glass
2. Plastic bottles
3. Batteries
4. Chemical office waste
5. Paper

*Appendix A.8. Environmental Commitment*

1. I really care about the environmental concern of my organization.
2. I would feel guilty about not supporting the environmental efforts of my organization.
3. The environmental concern of my organization means a lot to me.
4. I feel a sense of duty to support the environmental efforts of my organization.
5. I really feel as if my organization's environmental problems are my own.
6. I feel personally attached to the environmental concern of my organization.
7. I feel an obligation to support the environmental efforts of my organization.
8. I strongly value the environmental efforts of my organization.

*Appendix A.9. Green Lifestyle*

1. Recycle bottles, cans, or glass
2. Recycle newspaper
3. Compost garden waste
4. Take your own bags to the supermarket
5. Cut down on car use
6. Contribute money to environmental causes
7. Volunteer for an environmental group

*Appendix A.10. Environmental Consciousness*

1. The organization has clear and concrete environmental policies.

2. The managers in the organization are in charge of environmental policies.
3. I understand its environmental policies and environmental regulations in this organization.
4. I noticed that this organization implements regular environmental audits.

*Appendix A.11. Green Self-Efficacy*

1. We feel we can succeed in accomplishing environmental ideas.
2. We can achieve most environmental goals.
3. We feel competent to deal effectively with environmental tasks.
4. We can perform effectively on environmental missions.
5. We can overcome environmental problems.
6. We could find out creative solutions to environmental problems

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
