# Peer review of "An Investigation of Pro-Environmental Behaviour and Sustainable Development in Malaysia"

_sustainability, doi:10.3390/su12177083_

Round 1

Reviewer 1 Report

Any revision are required

Author Response

Dear reviewer,

We want to extend our appreciation to the reviewers for taking the time and effort necessary to provide such insightful guidance. We understand that peer reviewers are working in good faith and provide a crucial service to the advancement of knowledge and discovery worldwide.

We have carefully considered the comments offered by the three respected reviewers. Herein, we explain the revisions we have made to the paper based on those comments and recommendations.

In this context, we have modified the paper in response to the extensive and insightful reviewers’ comments. We believe that this revised version presents a more robust version of the paper, while maintaining a respectful balance between the Reviewers’ perspectives on the manuscript.

In the following section, we offer detailed responses to the Reviewers’ comments. After presenting explanation to reviewers, it is possible to find a ‘track change’ version of our paper, highlighted in turquoise green.

We shall be grateful if our article is considered favorably.

Thank you. 

Reviewer 2 Report

SUSTAINABILITY- 872612

An Investigation of the Pro-Environmental Behaviour and Sustainable Development in Malaysia

Review

The manuscript deals with the relevant research topic stressing the need to foster people personal sustainable behaviour starting from young generations. The manuscript aims to reveal the role of environmental commitment, environmental consciousness, green lifestyle, and green self-efficacy and its’ relationship with pro-environmental behaviour.

The paper is interesting and well structured. The discussion section is particularly valuable. Starting reading the article from this section might not raise any questions. However, some clarifications are desirable both in the abstract and hypothesis development section.

There is a lack of reasoning, why the paper focuses on previously (by other scholars) confirmed relationships between the constructs under investigation. Maybe there are any contradictory results found in literature, which can confirm the necessity of further research? On the other hand, perhaps it can be argued that the research of those relationships is important because of the different contexts examined in previous studies. The statement about the originality of the paper appears only in the discussion section; it would be worth to mention this in the abstract at least.

Please clarify, how was the sample size determined? Is it not too small for the proposed research framework? According to Benitez et al.[1] (2020) “PLS-PM can produce estimates even for very small sample sizes. However, as for other estimators, these estimates are generally less accurate than those obtained by a larger sample” (p.2).

In section 3.2. the questionnaire is described. It would be good to find those items in the Appendix.

Small remarks:

In rows 89-90, there is a following statement: “The study focuses on the actions and behaviours performed by the observed students distinguishing whether those action students’ initiatives in performing pro-environmental behaviour or they are employed within the green behaviour control of the training center”. It would be perfect to compare the real behavior of those students and their declaration regarding the sustainable consumption. However, it seems to me that it was simply a mistake regarding the word ‘observed’.

In rows 97-98 you state that “we then outline the research method, which is followed by the discussion of the analyses and results”. Maybe, first, we provide results and after that we discuss those results?

Row: 222 integration of “theory” or theories?

Please take into account those comments; otherwise please provide a response along with the final version of the paper. Thank you!

Good luck!

Best regards,

Reviewer

[1] Benitez, J.; Henseler, J.; Castillo, A. & Schuberth, F. How to perform and report an impactful analysis using partial least squares: Guidelines for confirmatory and explanatory IS research Information & Management, Elsevier BV, 2020, 2, 103168

Author Response

(The authors gave the same response as above.)

Reviewer 3 Report

It is advisable to state clearly the aim of this paper and justify the theoretical framework. If the authors rearrange and adopt a critical point of view when writing the theoretical framework, information will be then meaningful. There is a need of offering detailed insights regarding the main purpose of the paper and its contribution. Data gathering and data analysis can be reconsidered and discussed more comprehensively. The analysis strikes me as requiring a bit more depth and to clearly state the contribution that it makes, even though it aims to provide an overview of the topic. The presentation of the main constructs that anchor the argument could be strengthened. Several statements made in the paper are not supported by adequate empirical evidence or by making reference to relevant literature. There is some discussion of the limitations of the study however these are not considered in terms of the implications on the study findings. The conclusion should clarify the main contribution of the paper and the value added to the field.

More recent references from Scopus- or WoS-indexed journals are needed. Here are some research suggestions that may complement your approach (I am not the editor of these journals, member of the board, or reviewer):

Morgan, C. (2020). “Can Smart Cities Be Environmentally Sustainable? Urban Big Data Analytics and the Citizen-driven Internet of Things,” Geopolitics, History, and International Relations 12(1): 80–86.

Hodgkins, S. (2020). “Big Data-driven Decision-Making Processes for Environmentally Sustainable Urban Development: The Design, Planning, and Operation of Smart City Infrastructure,” Geopolitics, History, and International Relations 12(1): 87–93.

Gutberlet, Trish (2019). “Data-driven Smart Sustainable Cities: Highly Networked Urban Environments and Automated Algorithmic Decision-Making Processes,” Geopolitics, History, and International Relations 11(2): 55–61.

Jones, A., MuÈ™at, M., Corpodean, H., and Petris, G. (2020). “Sustainable Industrial Value Creation, Automated Production Systems, and Real-Time Sensor Networks in Big Data-driven Smart Manufacturing,” Journal of Self-Governance and Management Economics 8(2): 35–41.

Author Response

(The authors gave the same response as above.)

Round 2

Reviewer 3 Report

There are no indications that you have improved your manuscript according to any of my comments.

Round 3

Reviewer 3 Report

Changes according to my comments must be clearly indicated.

Author Response

Dear respected reviewer,

We want to extend our appreciation to you for taking the time and effort necessary to provide such insightful guidance.

As your suggestion, this paper need to proofread by native or professional. Thus, we have sent it to professional proofreader (Manuscript-Proofread). 

For introduction part,  we have revised and changed as your comments to provide sufficient background and relevant references. Next, the design of the study and appropriate method sections have clearly explained (Attached in Summary of Comments). In addition, the results and conclusion have been improved like your suggestion.

We thank you for your kind feedback and appreciate it very much.
